# The Zinc Finger Transcription Factor BbCmr1 Regulates Conidium Maturation in *Beauveria bassiana*

Jin-Feng Chen,[a,b] Jun-Jie Tan,[a] Jun-Yao Wang,[a] A-Jing Mao,[a] Xue-Ping Xu,[a] Yan Zhang,[a] Xue-li Zheng,[a] Yu Liu,[c] Dan jin,[a] Xian-Bi Li,[a] Yan-Hua Fan[a]

[a]Biotechnology Research Center, Southwest University, Beibei, Chongqing, People's Republic of China
[b]College of Biological and Chemical Engineering, Chongqing University of Education, Chongqing, People's Republic of China
[c]College of Biotechnology, Southwest University, Beibei, Chongqing, People's Republic of China

**ABSTRACT** The entomopathogenic fungus *Beauveria bassiana* is a typical filamentous fungus and has been used for pest biocontrol. Conidia are the main active agents of fungal pesticides; however, we know little about conidial developmental mechanisms and less about maturation mechanisms. We found that a $Zn_2Cys_6$ transcription factor of *B. bassiana* (named BbCmr1) was mainly expressed in late-stage conidia and was involved in conidium maturation regulation. Deletion of *Bbcmr1* impaired the conidial cell wall and resulted in a lower conidial germination rate under UV (UV), heat shock, $H_2O_2$, Congo red (CR) and SDS stresses compared to the wild type. Transcription levels of the genes associated with conidial wall components and trehalose synthase were significantly reduced in the Δ*Bbcmr1* mutant. Further analysis found that BbCmr1 functions by upregulating BbWetA, a well-known transcription factor in the central development of BrlA-AbaA-WetA. The expression of *Bbcmr1* was positively regulated by BbBrlA. These results indicated that BbCmr1 played important roles in conidium maturation by interacting with the central development pathway, which provided insight into the conidial development networks in *B. bassiana*.

**IMPORTANCE** Conidium maturation is a pivotal event in conidial development and affects fungal survival ability under various biotic/abiotic stresses. Although many transcription factors have been reported to regulate conidial development, we know little about the molecular mechanism of conidium maturation. Here, we demonstrated that the transcription factor BbCmr1 of *B. bassiana* was involved in conidium maturation, regulating cell wall structure, the expression of cell wall-related proteins, and trehalose synthesis. BbCmr1 orchestrated conidium maturation by interplaying with the central development pathway BrlA-AbaA-WetA. BbBrlA positively regulated the expression of *Bbcmr1*, and the latter positively regulated *BbwetA* expression, which forms a regulatory network mediating conidial development. This finding was critical to understand the molecular regulatory networks of conidial development in *B. bassiana* and provided avenues to engineer insect fungal pathogens with high-quality conidia.

**KEYWORDS** *Beauveria bassiana*, conidium maturation, BbCmr1, BbWetA

Address correspondence to Yan-Hua Fan, fyh@swu.edu.cn.

The authors declare no conflict of interest.

*B*eauveria bassiana, an invertebrate pathogenic fungi, is used worldwide for the biological control of pests and is perceived as a source of novel biocatalysts and metabolites (1–3). Fungal aerial conidia are produced asexually to escape harsh conditions, colonize new environmental niches, and recognize hosts. For fungal infection against insect hosts, conidia of *B. bassiana* adhere to the host cuticle and germinate to produce invading appressoria under appropriate temperature and humidity conditions (4, 5). Depending on mechanical pressure and cuticle-degrading hydrolases, fungi

penetrate the host cuticle and colonize the host hemolymph (6). Exhausted nutrition by fungi and released toxic secondary metabolites lead to the death of the host (7, 8). Subsequently, fungal hyphae grow outside cadavers and produce adequate conidia, which can cause a new round of infection with suitable hosts. As fungal germination and growth in nature or hosts are affected by environmental conditions, such as temperature, UV and humidity, the quality of aerial conidia greatly affects the survival and fitness of *B. bassiana*.

Conidial production and maturation in filamentous fungi are complex processes and contain multiple events. Perception of diverse cues results in phenotypic changes from vegetative mycelium to conidiophores, followed by the production of conidia. Generally, $\alpha$-glucans, $\beta$-glucans, chitin, and other polysaccharides are intricately assembled, cross-linked, and modified to form the fungal cell wall in response to environmental signals (9–11). In conidia, compatible solutes, including sugars, sugar alcohols, amino acids, betaine, and heat shock proteins, also accumulate to protect fungi against diverse stressors (12, 13). Disturbing the synthesis of $\beta$-1,3-glucan in *Metarhizium acridum* destroys cell wall integrity and decreases the hyperosmotic tolerance of mutants (14). When the gene encoding mannosyltransferase in *B. bassiana* is knocked out, the altered content of $\alpha$-glucan and chitin in the conidial wall reduces the hydrophobicity of the thin conidia (15). In addition, deletion of the hydrophobin genes *hyd1* and *hyd2* in *B. bassiana* significantly reduces conidial hydrophobicity, adhesion, and virulence (16). These studies indicate that conidial development, especially the events affecting cell wall structure and components, plays vital roles in the stress responses and pathogenesis of entomopathogenic fungi.

The molecular mechanisms of conidiogenesis in filamentous fungi are highly conserved and contain a central developmental pathway (CDP), BrlA-AbaA-WetA, and complex upstream/downstream regulatory networks (17). In *Aspergillus nidulans*, the signal cascade BrlA-AbaA-WetA is responsible for conidial development during the different stages through the orderly regulation of conidiation-specific genes (18–20). WetA is involved in conidiation as well as cell wall integrity, stress tolerance, and spore viability (18, 21, 22). In a wetA-defective mutant of *Aspergillus*, the expression levels of *brlA* and *abaA* are upregulated (23), while genes associated with the biosynthesis of trehalose, melanin, and hydrophobins are downregulated (24). In addition to WetA, the velvet regulator VosA of *Aspergillus flavus* is also involved in conidial development, regulating conidial trehalose biogenesis and stress tolerance (25). In *B. bassiana*, both WetA and VosA are responsible for conidiation as well as conidium maturation (26). However, it remains unknown whether there are other regulatory proteins involved in conidium maturation. Our previous study implied the functional role of BbSmr1 in conidiation and the production of secondary metabolites (8, 27). We compared the results of conidial RNA sequencing (RNA-seq) of Δ*Bbsmr1* with WT and identified a transcription factor, BbCmr1 (conidium maturation regulator 1), involved in regulating conidial development. Further study confirmed that BbCmr1 functions via *BbwetA* and is involved in the regulation of conidial cell wall structure, stress responses and trehalose synthesis.

## RESULTS

**Bbcmr1 is highly expressed at the late stage of conidiation in *B. bassiana*.** In our previous study, the transcription factor BbSmr1 was found to be a positive regulatory factor of *B. bassiana* conidiation. Deletion of *Bbsmr1* resulted in fewer conidia but increased conidial resistance against adverse conditions. Transcriptome analysis further identified several transcription factors with significantly changed expression levels in the *Bbsmr1* deletion mutant compared to the wild type (27). Among them, a putative GAL4-like Zn$_2$Cys$_6$ transcription factor (EJP63695.1) was upregulated in the *Bbsmr1*-deleted mutant and named BbCmr1. The genomic sequence of *Bbcmr1* was 2423 bp, containing two introns (57 bp and 69 bp). *Bbcmr1* cDNA encoded 764 amino acids, with a molecular size of 84.4 kDa and an isoelectric point of 7.86. BbCmr1 was potentially classified as a

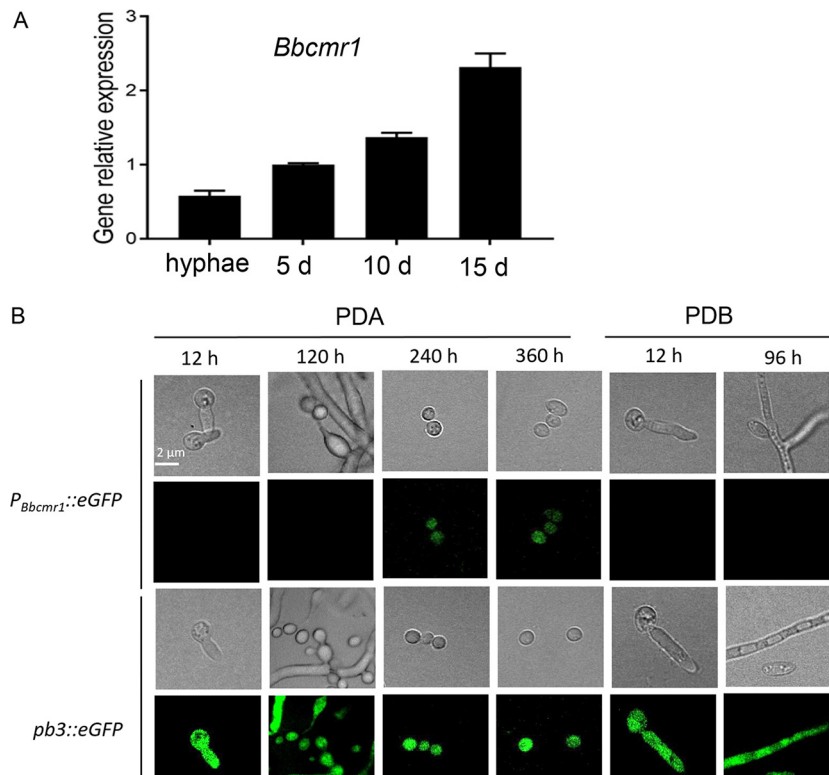

**FIG 1** Expressed profile of *Bbcmr1* in WT. (A) The relative transcript level of *Bbcmr1* in WT incubated in PDB for 4 d and on PDA for 5-15d. (B) DIC and fluorescence microscopic images of $P_{Bbcmr1}$::*eGFP* strain in liquid medium (PDB) for 12–96 h and on solid medium (PDA) for 12–360 h. GFP fluorescence was observed in germinated conidia, hyphae, blastospores, conidiophore, newly formed conidia, and mature conidia. *pb3*::*eGFP* strain with constructive promoter *pgpdA* (*pb3*) was used as a positive control (Scale bars, 2 $\mu$m).

fungal-specific transcription factor, with a GAL4-like $Zn_2Cys_6$ binuclear cluster DNA-binding domain at the N terminus and a nuclear localization signal RKRK (16–19) at the C terminus. Phylogenetic analysis revealed that BbCmr1 shared 94.9% sequence identity with the homologous protein from *Beauveria brongniartii* but only 40–70% identity with those from *Cordyceps militaris*, *Aspergillus flavus*, and other filamentous fungi (Fig. S1).

*Bbcmr1* expression was first detected by real-time PCR after culturing the *B. bassiana* wild-type strain on PDA for 5–15 d, a period spanning from conidiation initiation to conidium maturation. Generally, *Bbcmr1* exhibited a higher expression level on solid medium than on liquid medium. However, the expression of *Bbcmr1* on PDA gradually increased from 5 to 15 d, showing a 2.5-fold higher expression level on the 15th day than on the 5th day of culture from PDA (Fig. 1A). To further analyze the expression pattern of *Bbcmr1*, *eGFP* was fused with the *Bbcmr1* native promoter ($P_{Bbcmr1}$::*eGFP*). Strong GFP fluorescence was only observed in late-stage conidia (cultured on PDA for 240 h or 360 h) but not in hyphae, conidiophores or newly formed conidia (Fig. 1B). However, in the *pb3*::*eGFP* strain with a fusion of the constitutive promoter *pb3* and *eGFP*, intensive GFP fluorescence was observed in all cell types. These results indicated that *Bbcmr1* was mainly expressed in mature conidia and was possibly related to the conidium maturation process.

**Bbcmr1 is required for the viability of conidia under abiotic stresses.** To examine the contributions of BbCmr1 to conidium maturation, *Bbcmr1* deletion mutant isolates (Δ*Bbcmr1*) were constructed using homologous recombination and confirmed by PCR and RT–PCR analyses (Fig. S2). A complementary mutant (Δ*Bbcmr1RC*) was obtained by introducing the *Bbcmr1* gene containing a 2-kb promoter region and a 2295-bp downstream region into the Δ*Bbcmr1* mutant. Conidial germination percentages were

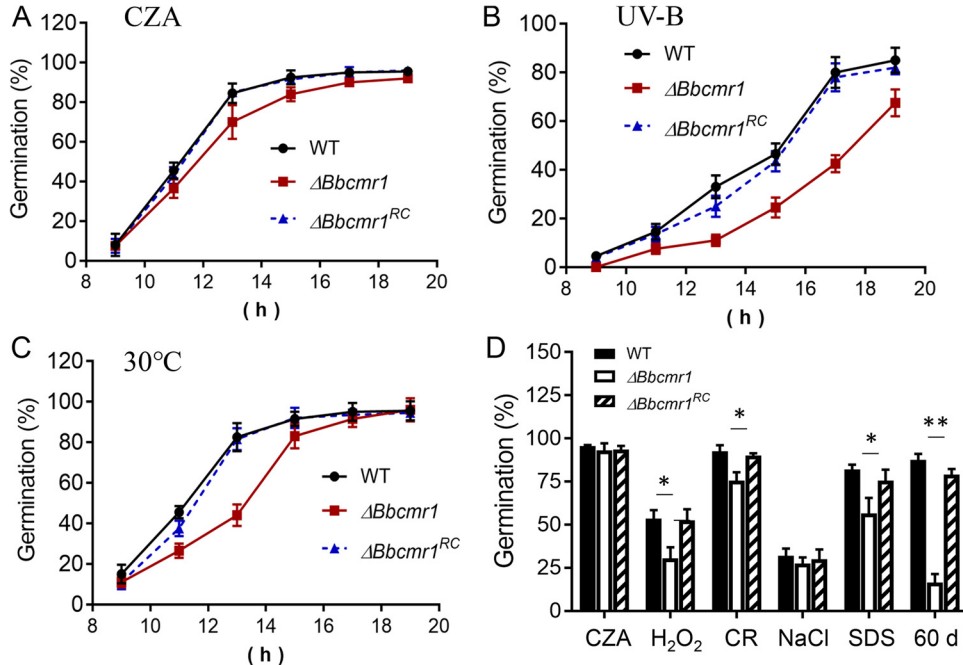

**FIG 2** Conidial germination rates of various strains under different stresses. The treatments, including standardized condition (A), UV-B irradiation (B), heat stress at 30°C (C), $H_2O_2$ (2 mM), SDS (0.2 mg/ml), Congo red (CR, 0.8 mg/ml), NaCl (1.2 M), and stored spores for 60 d at 26°C (D). *, $P < 0.05$, **, $P < 0.01$.

analyzed by inoculating conidial suspensions onto CZA plates (Czapek-Dox Agar) with different stressors, including $H_2O_2$ (2 mM), CR (0.8 mg/ml), SDS (0.2 mg/ml), or treatment with UV-B radiation and heat stress at 30°C. No significant difference in conidial germination was observed between WT and $\Delta Bbcmr1$ strains on CZA without any stressor, but a slight reduction did exist in the $Bbcmr1$ mutant (Fig. 2A). Once exposed to UV-B, $\Delta Bbcmr1$ conidia exhibited a lower germination rate than WT conidia (Fig. 2B), with a 20% to 46.9% reduction from 15 to 17 h postinoculation. The time to reach 50% germination ($GT_{50}$) for $\Delta Bbcmr1$ conidia with UV-B treatment was 2 h longer than that for WT conidia (17.5 versus 15.5 h, $P < 0.05$). Similarly, the conidia of $\Delta Bbcmr1$ were sensitive to heat stress. After culturing for 13 h at 30°C, approximately 80% of WT and complementary conidia had germinated, but only 50% germinated conidia were observed in $\Delta Bbcmr1$, with a 13.0% increase in $GT_{50}$ compared with WT ($\Delta Bbcmr1\_13$ h versus WT_11.5 h) ($P < 0.05$) (Fig. 2C). In addition, when supplied with $H_2O_2$ (2 mM), CR (0.8 mg/ml), or SDS (0.2 mg/ml), the conidial germination rates of $\Delta Bbcmr1$ (24 h postinoculation) were decreased by 23.0%, 17.0%, and 20.4%, respectively, compared to WT ($P < 0.05$) (Fig. 2D). Although wild-type and mutant conidia (15 d old) exhibited similar germination rates on CZA without any stressors, long-term storage of mutant conidia under a dry environment (60 d) resulted in a 71.0% reduction in germination rate compared to wild-type conidia (Fig. 2D). The above altered phenotypes were well restored in complementation strains.

**Deletion of *Bbcmr1* changes the cell wall and intracellular components of conidia.** We examined the effects of BbCmr1 on conidial morphology and surface features, which are important indicators of conidium maturation. TEM results indicated that the cell wall of $\Delta Bbcmr1$ conidia had weak electron density and was thinner than that of wild-type conidia (Fig. 3A). Sample preparation for TEM also resulted in irregular cell shapes in the $Bbcmr1$ deletion strain, indicating a change in rigidity (Fig. 3A). Surface carbohydrate epitopes were examined by three fluorescent lectins, WGA, PNA and ConA, which recognized $\beta$-GlcNAc and sialic acid residues, $\beta$-galactose, and $\alpha$-glucose and $\alpha$-N-acetylglucosamine (GlcNAc), respectively. Compared to wild-type and complementary strain conidia, three tested lectins exhibited weak binding ability to

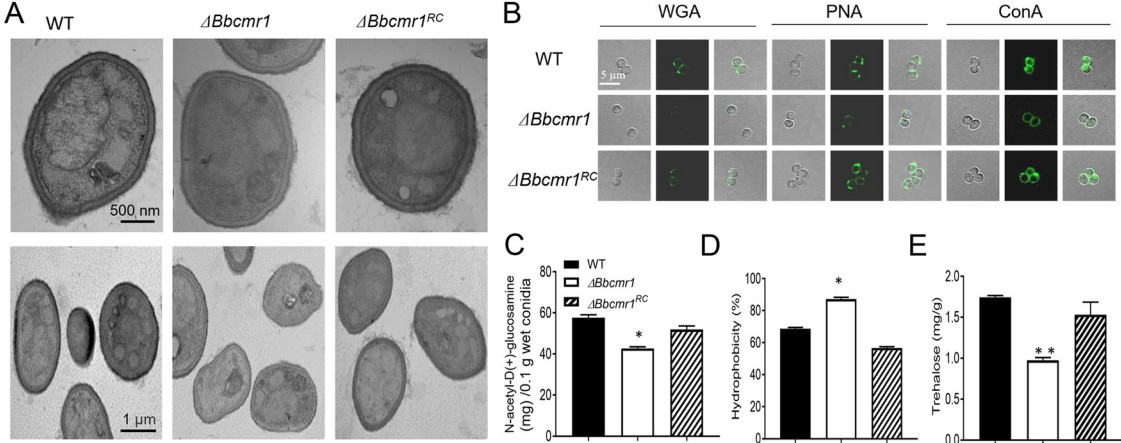

**FIG 3** BbCmr1 affects cell wall properties of conidia. (A) Representative TEM images of conidia. (B) Fluorescence images of lecting binding profiles of aerial conidia. Fluorescent lectins WGA, PNA and ConA were specific for β-GluNAc and sialic acid residues, β-galactose, and α-Glc/α-GlcNA respectively (scale bar, 5 μm). (C) chitin content, (D) hydrophobicity, and (E) intracellular trehalose accumulation from 15-day-old aerial conidia incubated on 1/4SDAY. *, $P < 0.05$, **, $P < 0.01$.

Δ*Bbcmr1* conidia (Fig. 3B). Deletion of *Bbcmr1* also resulted in a 25.9% decrease in chitin content ($P < 0.05$) and a 27.0% increase in conidial hydrophobicity ($P < 0.05$) (Fig. 3C and D), reflecting defective properties of the cell wall. Additionally, the content of intracellular trehalose in Δ*Bbcmr1* conidia, which is important for the stress response, decreased to 56% of that in WT conidia ($P < 0.01$) (Fig. 3E).

**BbCmr1 regulates the expression of genes involved in conidium maturation.** To further assess the potential functions of BbCmr1 in conidial development, a global analysis of the transcriptomes of Δ*Bbcmr1* and wild-type strains grown on PDA for 10 d was performed. A total of 1061 genes were differentially expressed in Δ*Bbcmr1* compared to the WT strain, including 962 downregulated genes and 99 upregulated genes (Fig. 4A). Functional category analysis using Gene Ontology (GO) terms showed that differentially expressed genes belonged to the functions related to metabolism and transport pathways of sugar, lipids, protein, and biosynthetic pathways of cell wall structure (Fig. 4B and C).

Proteins associated with structural wall components were responsible for fungal development, stress responses and virulence (28). The regulatory role of BbCmr1 in the synthesis of conidial cell wall components was analyzed by real-time PCR. The mRNA levels of 10 genes encoding conidial cell wall components (28), including chitinase-like protein (tag loci: BBA_06297), alpha-glucosidase b (tag loci: BBA_01877), catalase-peroxidase (tag loci: BBA_09760), and Rds1 (tag loci: BBA_07526, regulated by different signals), were tested by RT–PCR in the *Bbcmr1* mutant and wild type. Most gene transcription in Δ*Bbcmr1* cultivated for 10 d or 15 d was significantly depressed by 10–90% compared with WT (Fig. 5A). To further confirm the results, the red fluorescent protein (mCherry) tagged-BbRds1 (Rds1::mCherry) was expressed in the Δ*Bbcmr1* strain and the wild-type strain. As shown in Fig. 5B, the fusion protein was specifically distributed in the cell wall of the mature conidia (240 h-360 h) but gradually disappeared when the conidia began to germinate. Compared to the wild type, however, the fluorescence signal of Rds1::mCherry in the Δ*Bbcmr1* strain was obviously weaker in mature conidia (Fig. 5B). At the same time, regulatory genes in the coordination of conidial development, such as *BbwetA*, *Bbsmr1* and *BbbrlA* were also involved in the differentially expressed genes (DEGs) in the transcriptomes.

**BbCmr1 is required for the expression of BbwetA.** BbWetA is a regulator of conidiation capacity and conidial quality in *B. bassiana* (26). To clarify the relationship between *Bbcmr1* and *BbwetA*, the transcription level of *BbwetA* was detected in 5-, 10-, 15-, and 20-day-old conidia of the WT and Δ*Bbcmr1* strains. During the period of conidial development, the *BbwetA* transcript in the wild type gradually increased from 5–20 d

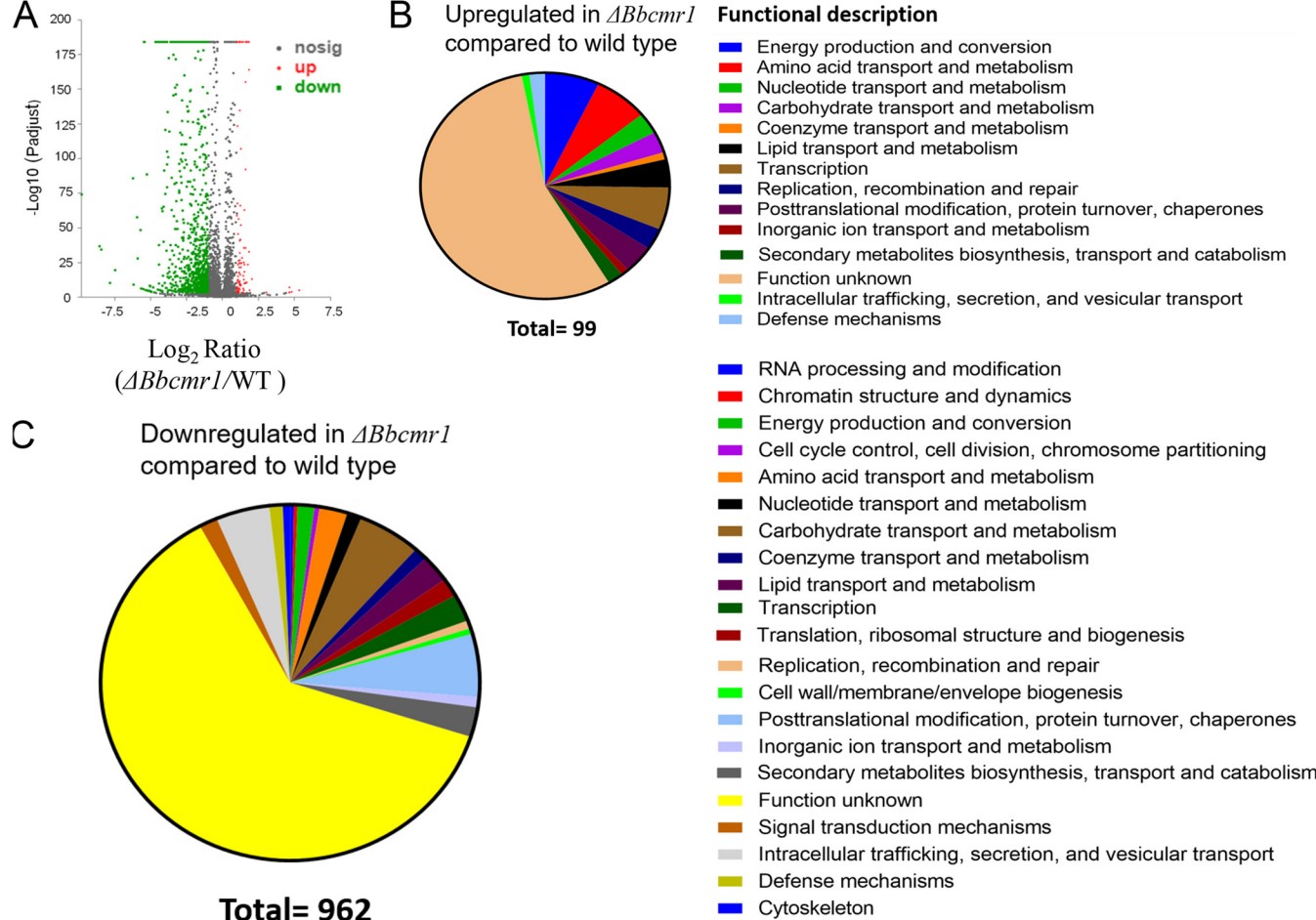

**FIG 4** Transcriptional analyses of genes influenced by *Bbcmr1*. (A) Cluster analysis of differentially expressed genes (DEGs) in the transcriptomes from the 10 d-old PDA cultures of ΔBbcmr1 and WT strains. Distributions of log$_2$ ratios (≥1 or≤ −1) and adjusted *P* values ($P_{adjust}$ <0.05) for 1061 DEGs identified from ΔBbcmr1. Upregulated (B) and downregulated (C) genes in ΔBbcmr1 were categorized according to putative functions gathered from Fungal Genome Database (FunCat).

(Fig. 6A). In the ΔBbcmr1 strain, the expression of *BbwetA* exhibited a similar trend as in the wild type during conidial development; however, the expression level decreased by 60%–80% compared with those at the same time points in the wild type (Fig. 6A). A yeast one-hybrid assay proved that BbCmr1 was able to bind the *BbwetA* promoter and regulate gene expression (Fig. 6B).

**Overexpression of *BbwetA* in *Δ Bbcmr1* rescues conidial development deficiency.** To further clarify the relationship between *Bbcmr1* and *BbwetA*, *BbwetA* and *Bbcmr1* were overexpressed in ΔBbcmr1 and ΔBbwetA, respectively (i.e., ΔBbcmr1/BbwetA$^{OE}$ and ΔBbwetA/Bbcmr1$^{OE}$). Several conidium maturation-related genes were detected. In the ΔBbcmr1 and ΔBbwetA strains (Fig. S3), conidial cell wall component genes (tag loci: BBA_07526; BBA_03717; BBA_09760) and the trehalose synthase gene (tag loci: BBA_02123) were downregulated compared to those in the wild-type strain (Fig. 7). Overexpression of *BbwetA* in ΔBbcmr1 (Fig. S4) resulted in a significant upregulation of these genes, but overexpression of *Bbcmr1* in ΔBbwetA did not change their expression level compared to ΔBbwetA (Fig. 7). These results indicated that BbCmr1 regulated co-nidium maturation-related gene expression via BbWetA. Conidial germination of these strains was analyzed on CZA supplied with 0.2 mg/ml SDS. Consistent with the gene expression results, ΔBbwetA and ΔBbcmr1 were equally more sensitive to SDS, and overexpression of *BbwetA* in ΔBbcmr1 rescued the resistance level similar to WT, but overexpression of *Bbcmr1* in ΔBbwetA strains did not restore the resistance level (Fig. 8A and B).

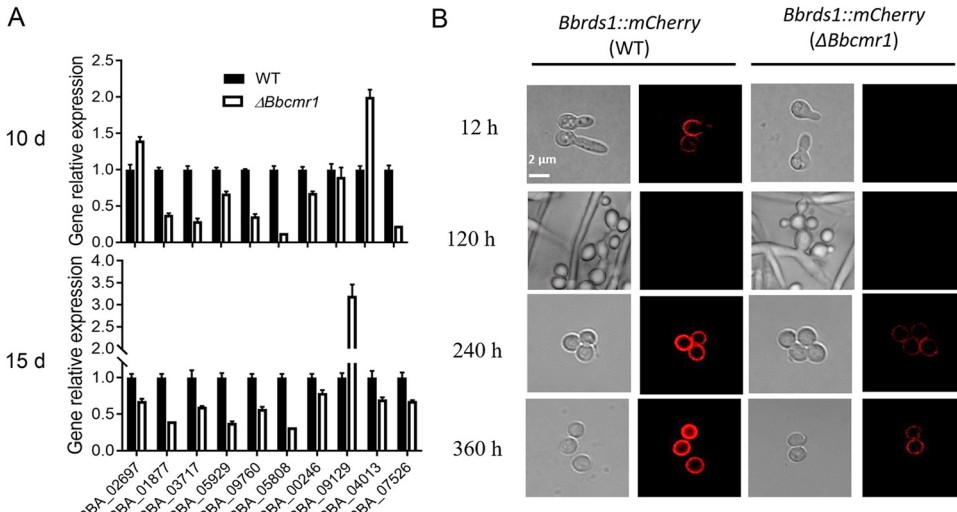

**FIG 5** (A) Relative transcript levels of the conidial wall component genes in Δ*Bbcmr1* strain culture grown for 10 d or 15 d on PDA under 26℃. (B) Subcellular localization of BbRds1::mCherry fusion proteins expressed in *Bbcmr1* deletion and WT strains. Fungal strains were cultured on the solid medium PDA for 12–360 h. Red fluorescent signals in different cell types were observed. (Scale bar, 2 μm).

**BbBrlA positively regulates *Bbcmr1* expression.** To analyze the genetic relationship between *Bbcmr1* and other conidiation-related genes, the mRNA levels of *Bbsmr1*, *BbbrlA* and *BbabaA* in various strains were detected by real-time PCR. Compared to the wild type, deletion of *Bbcmr1* resulted in significant increases in *Bbsmr1* and *BbbrlA* transcripts by 1.7- and 2.7-fold, respectively, but no significant effect on *BbabaA* (Fig. 9A). Conversely, almost no expression of *Bbcmr1* was detected in the *BbbrlA* deletion mutant, but the overexpression of *BbbrlA* (Δ*Bbsmr1/BbbrlA^OE*) improved *Bbcmr1* expression, indicating that BbBrlA is a positive regulator of *Bbcmr1* expression. However, BbSmr1, an upstream positive regulator of *BbbrlA*, negatively regulated *Bbcmr1* expression, with a higher expression level of *Bbcmr1* in the *Bbsmr1* deletion mutant (Fig. 9B).

BrlA is required for conidiophore development and binds the CCCCT motif (27). This motif was found in the *Bbcmr1* promoter, implying that BbBrlA could directly regulate *Bbcmr1* expression. A fragment of the *Bbcmr1* promoter region containing the putative BbBrlA response element (642 bp to 692 bp upstream of ATG) was chosen as a probe to conduct EMSA (Electrophoretic Mobility Shift Assay). GST-fused full-length BbBrlA was expressed and purified from *Escherichia coli*. EMSA showed that BbBrlA could directly bind the promoter region of *Bbcmr1*, but the shifted band disappeared when

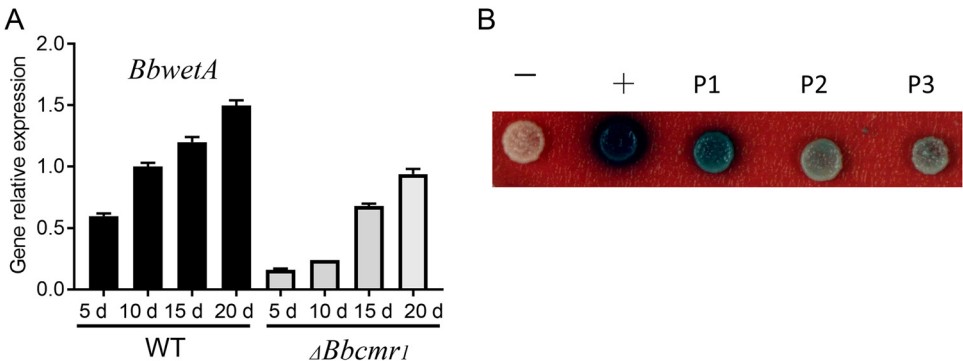

**FIG 6** (A) *BbwetA* expressed in the conidia of fungi. Relative transcript levels of *BbwetA* in Δ*Bbcmr1* and wide-type strains grown on 1/4SDAY for 5–20 d at 26℃. (B) Yeast one-hybrid analysis of the interaction of BbCmr1 and the *BbwetA* promoter. The promoter of *BbwetA* was divided into P1 (−1049 ~ −1500), P2 (−548 ~ −1048), and P3 (−1 ~ −547).

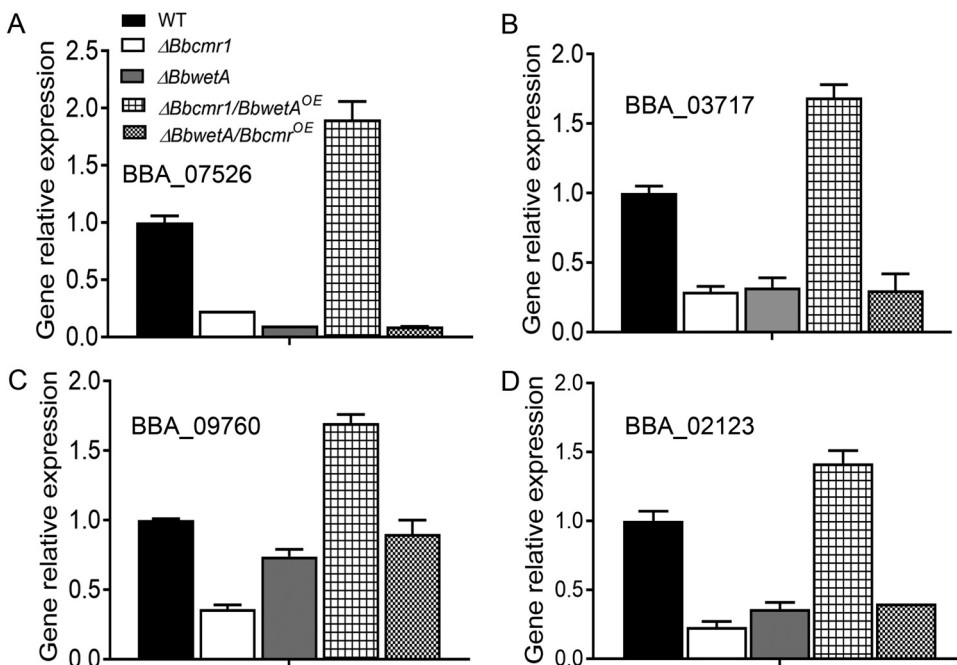

**FIG 7** The relative expressions of conidial wall component genes (A-C) and intracellular trehalose synthase genes (D) in Δ*Bbcmr1/BbwetA*$^{OE}$ and Δ*BbwetA/Bbcmr1*$^{OE}$ grown on 1/4SDAY for 10 d.

the competitive probe was added to the assay (Fig. 10A). A yeast one-hybrid assay also confirmed the EMSA results (Fig. 10B), suggesting that BbBrlA was able to recognize *Bbcmr1* promoter regions and activate their expression.

To perform EMSA, BbCmr1 was heterologously expressed and purified from *Escherichia coli* (Fig. S5). As deletion of *Bbcmr1* changed the expression of some conidial cell wall-related genes, we hypothesized that BbCmr1 could bind directly to the promoters of these genes. However, using several predicted conserved motifs from the promoters of these genes, no obvious binding was observed (data not shown). Expression analysis results showed that BbCmr1 had a repressive effect on *BbbrlA*. We tested several DNA fragments from the *BbbrlA* promoter, and a fragment containing AAAAGAAA (A4GA3) motifs exhibited typical protein concentration-dependent binding with BbCmr1 (Fig. 10C). Adding excess competitors (200- to 800-fold) into the assay resulted in decreased intensity of the DNA-protein complex (Fig. 10C). The EMSA results indicated that BbCmr1 could bind the promoter of *BbbrlA*. A yeast one-hybrid

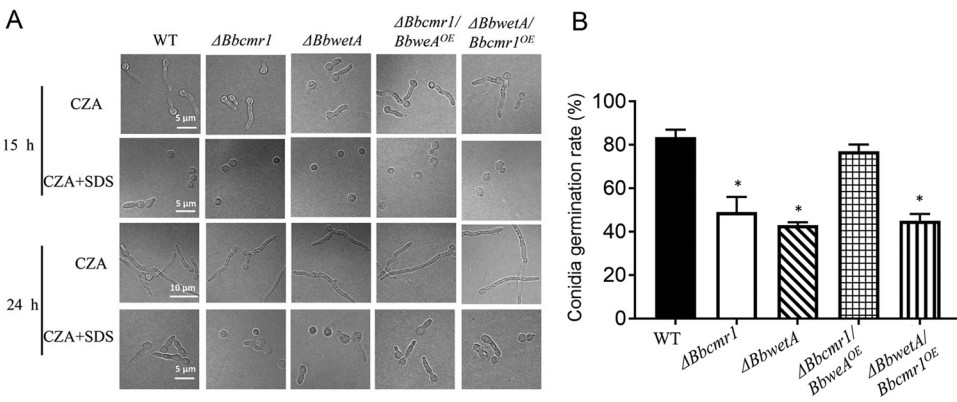

**FIG 8** Conidial germinations of the fungal strains under SDS stress. (A) Conidial suspensions (100 μl, 1×10⁸ conidia/ml) were sprayed onto CZA plates (90 mm) containing SDS (0.2 mg/ml) and cultivated for 24 h. Germinated conidia were microscopically observed (A) and the germination rates were analyzed (B). *, P < 0.05. (scale bars, 5 μm).

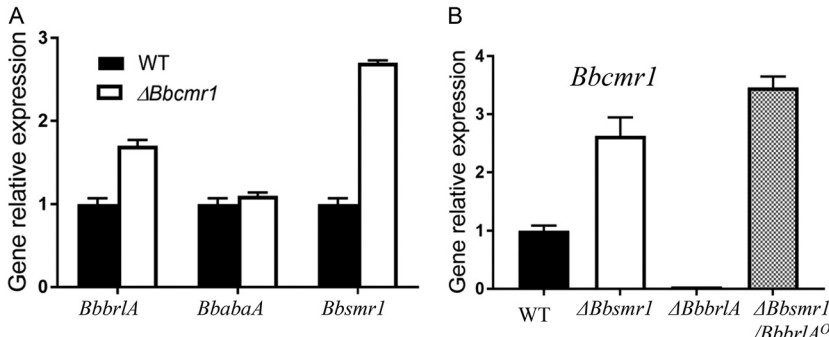

**FIG 9** (A) Relative transcript levels of conidiation-related genes from 10-day-old cultures of *Bbcmr1* deletion mutant on PDA. (B) Relative transcript level of *Bbcmr1* in Δ*Bbsmr1*, Δ*BbbrlA*, and Δ*Bbsmr1/BbbrlA*<sup>OE</sup> (overexpression *BbbrlA* in Δ*Bbsmr1* background) strains grown on PDA for 10 d, respectively.

assay also confirmed the interaction between BbCmr1 and the promoter of *BbbrlA* (Fig. 10D). Further analysis showed that BbCmr1 could bind the A4GA3 sequence but not the A4G2A2 motif (Fig. 10E).

Combining these results, we summarized a regulatory cascade mediating conidium maturation in *B. bassiana* (Fig. 11).

## DISCUSSION

**BbCmr1 is a vital regulator of conidiation.** $Zn_2Cys_6$ transcription factor (TF) genes are unique to fungi and are involved in the processes of fungal development, pathoge-

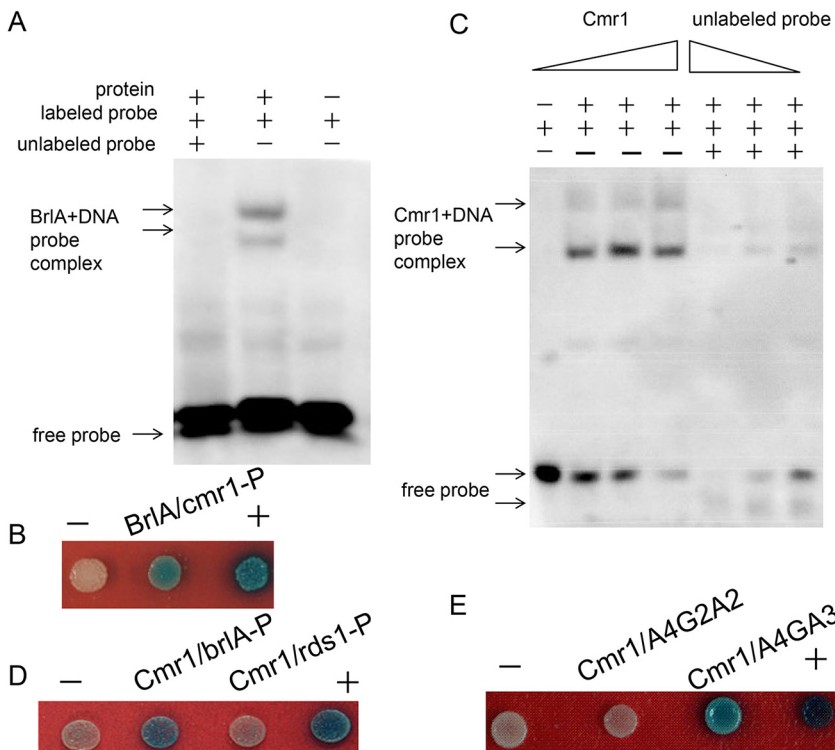

**FIG 10** (A) EMSA of the interaction between BbBrlA and promoter of *Bbcmr1*. The promoter region of *Bbcmr1* (642 bp to 692 bp upstream of ATG) was used as labeled probe, and unlabeled probe was added in a 200-fold excess. (B) Yeast one-hybrid analysis of BrlA binding the promoter of *Bbcmr1*. (C) EMSA of the binding activity of BbCmr1 with the promoter of *BbbrlA* (586 bp to 637 bp upstream of ATG). Each lane contained 10 ng labeled probe, purified protein (0.5 ~ 2 μg) or purified protein (2 μg) and unlabeled probe (200 ~ 800–fold excess) were added in reactions. (D) Yeast one-hybrid analysis for the interaction between BbCmr1 and the promoter of *BbbrlA*. (E) A4GA3 was the possible binding site of BbCmr1.

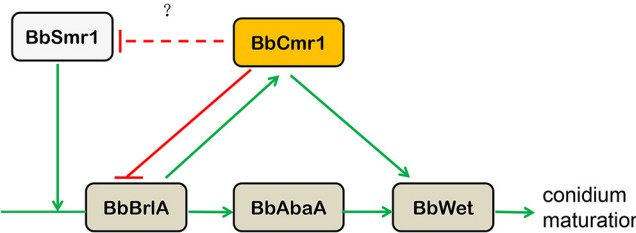

**FIG 11** Regulatory networks of conidium maturation. BbCmr1 was responsible for trehalose accumulation and cell wall integrity via BbWetA. BbWetA and BbVosA were both required for conidium maturation. Green arrows indicated positive regulation, and red lines indicated negative regulation, dotted lines represented the uncertain control relationship.

nesis and stress tolerance (29–31). Over 40 putative $Zn_2Cys_6$ transcription factors have been found in the *B. bassiana* genome. However, only a few have been characterized (32). Recently, the $Zn_2Cys_6$ TF BbTpc1 has been found to regulate fungal development, chitin synthesis and biological control potential (33). The present study showed that *Bbcmr1*, which encodes a GAL4-like $Zn_2Cys_6$ transcription factor, was mainly expressed in late-stage conidia and regulated conidium maturation.

The conidiation event was temporally and spatially controlled by the developmental regulatory cascade BrlA-AbaA-WetA (20, 27, 34). The initiation and maturation of conidia are two distinct processes during conidial development in filamentous fungi and are regulated by different transcription factors (35). BrlA is a key transcription factor mediating the initiation of conidial development, and WetA and VosA are involved in conidium maturation (26, 36). In *Aspergillus* spp., AbaA activates WetA during the late phase of conidiation, and the latter regulates the events of conidium maturation, including trehalose synthesis, conidial wall formation, and secondary metabolism production (18, 24, 37). The velvet domain protein VosA is also required for conidium maturation (38, 39). Our results indicated that BbCmr1 is involved in the regulation of the BrlA-AbaA-WetA pathway. First, BbBrlA could bind the CCCCT site in the *Bbcmr1* promoter and activate its expression. Conversely, BbCmr1 was able to bind the promoter of *BbbrlA* and inhibit its expression. Second, BbCmr1 could promote the expression of *BbwetA* and be involved in conidium maturation regulation. In addition, our results also indicated an interplay between BbCmr1 and BbSmr1, a vital regulator of conidial development upstream of BbBrlA (27). BbCmr1 was able to inhibit *Bbsmr1* expression, and the latter also depressed *Bbcmr1* expression, suggesting that both factors were important components in the control of late-stage conidial development events. These results indicated that conidial development regulation is complex and that many transcription factors are involved in this process. Some regulatory proteins interacting with or upstream of the central pathway have been characterized, such as Flug, FlbB, and FlbC (40–42). However, there are few reports about the regulation of conidium maturation. Our study identified a new regulatory protein involved in conidium maturation and provided insight into the regulatory network of conidial development in *B. bassiana*.

**BbCmr1 contributes to cell wall integrity and multistress tolerance.** Cell wall integrity plays an important role in fungal stress tolerance. In *B. bassiana*, inactivation of $\alpha$-1,2-mannosyltransferases alters the cell wall components, resulting in thinner cell walls and mutants that are more sensitive to multistress (15). Deletion of the polyketide gene *BbpksP* also impairs conidial cell wall structure and reduces the UV-B tolerance of conidia (43). It was observed that *Bbcmr1* mutant conidia were more sensitive to SDS and UV-B irradiation than WT conidia. The significant changes in stress tolerance were likely attributable to changes in the composition of the conidial cell wall. We observed decreased contents of surface carbohydrate epitopes and chitin in $\Delta Bbcmr1$. In addition, impaired cell wall structure and reduced expression of cell wall genes were also observed in the *Bbcmr1* mutant.

The intracellular accumulation of trehalose protects conidia against unfavorable environments, particularly against thermal stress and long-term viability. Trehalose accumulates during conidial development, but this content drops rapidly when the conidia germinate under suitable environmental conditions (44). Blocking trehalose accumulation in *A. fumigatus* reduces spore viability and seriously impairs subsequent events, including germination, growth, and sporulation, after exposure to heat (50°C) (45). In *A. fumigatus*, the lack of trehalose in Δ*mybA* conidia results in the rapid loss of viability (46). Our results found that genes responsible for intracellular trehalose production were downregulated in Δ*Bbcmr1*, and the content of intracellular trehalose was also reduced. We supposed that the decreased viability and significantly increased susceptibility of Δ*Bbcmr1* conidia to stressors could be partially attributed to trehalose changes, as observed in other filamentous fungi (45, 47, 48).

**BbCmr1 promotes the maturation of spores through BbWetA.** BbCmr1 regulates conidium maturation in the late phase of conidiation, similar to the functional characteristics of WetA reported in *B. bassiana* (26). Δ*Bbcmr1* and Δ*BbwetA* showed similar defects in conidial tolerance to UV-B irradiation and cell wall perturbation SDS. Impaired cell walls, decreased intracellular trehalose content, and decreased viability were also observed in both mutant strains. When *BbwetA* was overexpressed in the Δ*Bbcmr1* mutant, the expression of conidial cell wall-specific genes was significantly upregulated, and tolerance to SDS stress was restored. In contrast, overexpression of the *Bbcmr1* gene in Δ*BbwetA* did not significantly upregulate the mRNA levels of cell wall genes and trehalose synthesis genes, indicating that BbCmr1 regulated the expression of genes associated with conidial maturation through *BbwetA*. Although BbVosA was also involved in conidium maturation, our present data failed to verify its relationship with BbCmr1 due to the complex regulatory networks during conidial development.

In summary, the transcription factor BbCmr1 plays a vital role in conidium maturation in *B. bassiana*, mediating many aspects of conidial biology involved in cell wall integrity, stress tolerance, and spore viability. BbCmr1 was able to coordinate conidial formation and conidium maturation by repressing the expression of *BbbrlA* but activating *BbwetA* expression. Future research concerning the relationship of BbCmr1 with other regulatory proteins will further elucidate the genetic regulatory networks of conidial development in *B. bassiana*.

## MATERIALS AND METHODS

**Strains and culture conditions.** *B. bassiana* strains (CGMCC7.34, China General Microbiological Culture Collection Center) were maintained on potato dextrose broth/agar (PDB/PDA, BD, Difco), 1/4 Sabouraud dextrose broth/agar (1/4 SDAY/1/4 SDB, 1% glucose, 0.25% peptone, 0.5% yeast extract, optional 1.5% agar), and Czapek-Dox broth/agar (CZB/CZA, BD, Difco). *Escherichia coli* DH5α used for plasmid propagation was cultured in Luria-Bertani (LB) medium supplemented with ampicillin (100 μg/ml) or kanamycin (50 μg/ml). For fungal transformation, *Agrobacterium tumefaciens* AGL-1 was cultured in yeast extract broth (1% peptone, 0.5% yeast extract, 0.05% MgSO$_4$.7H$_2$O).

**Construction of disruption strains.** The *Bbcmr1* and *BbwetA* genes (tag loci: BBA_07339 and BBA_06126) were individually disrupted from the wild type using a homologous recombination method. For construction of the *Bbcmr1* deletion vector, 2200-bp upstream fragments and 1600-bp downstream fragments were amplified with *Bbcmr1-F1/F2* and *Bbcmr1-R1/R2* primer pairs. A 1900-bp fragment within the *Bbcmr1* gene was homogenously replaced by a *bar* gene cassette via the *Agrobacterium*-mediated method (49). Transformants were screened for phosphinothricin resistance and were verified by PCR and RT–PCR using the primers listed in Table S1. Likewise, *BbwetA* mutants were generated with a sulfonylurea resistance cassette.

**Construction of the complementation strain.** A fragment containing the entire coding sequence of *Bbcmr1* (2295 bp) under the control of a native promoter (~2.0 kb) was cloned into pK2-sur at the *Eco*RI site via homologous recombination. The complementation vector was ectopically transformed into the Δ*Bbcmr1* strain via *A. tumefaciens*-mediated fungal transformation. The integrity of the insert was verified by PCR using primers *ptrpc-F/OEcmr1-R* and RT–PCR.

**Construction of the constitutive expression strains.** The coding sequences of *Bbcmr1* and *BbwetA* (2295 bp and 1920 bp, respectively) were amplified from *B. bassiana* cDNA with individual primer pairs. The PCR products were cloned into the EcoRI sites in pK2-sur (for overexpression *BbwetA* in Δ*Bbcmr1*) and pK2-bar (for overexpression *Bbcmr1* in Δ*BbwetA*) downstream of the constitutive *pgpdA* promoter. The overexpression vectors were ectopically transformed into the mutant strain. The putative transformants were verified by PCR and real-time PCR.

***eGFP*-promoter reporter analyses and BbRds1 localization.** The ~2.0-kb promoter region of *Bbcmr1* was amplified from *B. bassiana* genomic DNA and cloned into the pK2-sur vector at the 5′ end of the *eGFP* coding sequence, and the resulting construct was introduced into *B. bassiana* WT by *A. tumefaciens*-mediated transformation. GFP fluorescence signals were observed in diverse cell types using an Olympus microscope (FV1000, Japan). To observe the expression and localization of BbRds1, the ORF of *Bbrds1* (1410 bp) without the stop codon together with the putative native promoter region (1500 bp) was fused with the *mCherry* coding sequence by PCR with *Bbrds1-F/mCherry-R* primers. The final PCR product was cloned at the EcoRI site in the pK2-sur vector. This resulting plasmid was transformed into the WT or ΔBbcmr1 strain for fluorescent signal observation.

**Determination of the conidial response to various stresses.** Conidial tolerance to different types of chemical stressors was performed as described (26). One hundred microliter aliquots of conidial suspension ($1 \times 10^8$ conidia/ml) were spread onto CZA medium supplemented with 0.2 mg/ml SDS, 1.2 M NaCl, 2 mM $H_2O_2$, or 0.8 mg/ml Congo red (CR) and cultured at 26°C for 24 h. Conidial germination on medium was observed under light microscopy. To detect UV-B sensitivity, CZA plates inoculated with conidia were exposed to UV-B irradiation at wavelengths of 320–400 nm at a dose of 190 mJ/cm$^2$ for 2 s. To determine the conidial tolerance to high temperature, conidial samples were cultured at 30°C. To assess the viability of the conidia, 60-day-old conidia were cultured on CZA for 15 h, and the germination percentage was determined.

**Analyses of conidial properties.** For conidial characteristic analyses, *B. bassiana* conidia were collected from 15-day-old strains cultured on PDA. Transmission electron microscopy (TEM) observations were performed as described in reference (50). Briefly, fungal conidia were suspended in sterile 0.05% (vol/vol) Tween 80, collected by centrifugation (12,000 rpm; 10 min), and fixed in 2.5% glutaraldehyde solution (1 ml of 25% glutaraldehyde reagent, 4 ml of $H_2O$ and 5 ml of 0.2 M pH 7.4 phosphate buffer) overnight at 4°C for TEM observation. Carbohydrate epitopes on the surfaces of conidia were detected following the method of Wanchoo et al. (50). Conidial hydrophobicity and the chitin content in the conidial wall were also analyzed as previously described (51–53).

**Assays for intracellular trehalose content.** Trehalose content was extracted as described previously (26). Aliquots (1 g) of 10-day-old 1/4 SDAY cultures containing mycelia and conidia were ground with liquid nitrogen and transferred into test tubes. After adding 1 ml of ddH$_2$O, the samples were incubated at 98°C for 6 h and centrifuged for 20 min at 12,000 rpm. The content of trehalose in the supernatant was measured with trehalose content assay kits according to the manufacturer's instructions (Grace Biological Technology, Suzhou, China).

**Real-time quantitative PCR.** Fungal strains were cultured on PDA medium for 5–20 d. Total conidial RNA was extracted using the RNeasy Plant Miniprep kit (Qiagen, China) following the manufacturer's instructions and reverse transcribed using a Reverse transcription kit (TaKaRa, China). RT–PCR was performed with the iCycler iQ multicolor real-time PCR detection system with SYBR green (Bio–Rad). Each reaction was run in triplicate. The relative transcript levels of target genes were normalized to *actin* (GenBank accession no. HQ232398). For transcriptome sequencing, 10-day-old cultures were collected from PDA plates. Total RNA extraction was used for RNA sequencing (Meiji Biological Company, Shanghai, China).

**Yeast one-hybrid assay.** To perform a yeast one-hybrid assay, the *Bbcmr1* ORF (2295 bp) was amplified with the primer pair *Bbcmr1-42AD-F/R* using *B. bassiana* cDNA as a template and cloned into pB42AD to obtain pB42AD-*Bbcmr1* (effector). The *BbbrlA* promoter fragments (−586 ~ −637), (A4GA3)$_3$, and (A4G2A2)$_3$ were artificially synthesized and inserted into pLacZi to construct reporter vectors. The effectors and reporters were cotransferred into the *EGY48* yeast strain to confirm interactions in yeast cells. Transformants were screened on SD/-Ura/-Trp selective medium at 30°C for 3–5 d, and positive colonies were transferred onto Minimal Synthetically Defined Medium Ga/Raf supplemented with X-gal (80 mg/liter) for color development, using empty pB42AD and pLacZi as negative controls. Similarly, a one-hybrid technique was used to determine the interaction between BbBrlA and the *Bbcmr1* promoter (−642 ~-692).

**Expression and purification of BbCmr1.** To express GST-tagged proteins in *E. coli*, the *Bbcmr1*-encoding sequence was amplified and fused with the 3 end of *GST* in the pGEX-6p vector. The resulting vectors were introduced into *E. coli* BL21(DE3) for protein expression. Isopropyl $\beta$-d-thiogalactopyranoside (IPTG, 0.1 mM) was added to induce protein expression for 20 h at 20°C. Proteins were purified using the MagneGST Protein Purification System (Promega, USA) and verified by SDS–PAGE (Fig. S5).

**EMSA.** For EMSAs, biotin-labeled DNA fragments containing putative binding sites (−586 ~ −637 in the *BbbrlA* promoter, −642 ~ −692 in the *Bbcmr1* promoter) were synthesized (Huada Biological Technology, Beijing, China). EMSAs were conducted using the LightShift Chemiluminescent EMSA kit (Thermo Fischer Scientific, USA) following the manufacturer's instructions. In competition assays, the unlabeled probe was added in a 200–800-fold excess.

**Data availability.** The sequence data have been deposited in the Sequence Read Archive (SRA) under the accession PRJNA777036.

## SUPPLEMENTAL MATERIAL

Supplemental material is available online only.

**SUPPLEMENTAL FILE 1**, PDF file, 0.6 MB.

## ACKNOWLEDGMENTS

This study was funded by the grants from the National Natural Science Foundation of China (31570137 & 31770158), the Fundamental Research Funds for the Central Universities (XDJK2018AA006), the Chongqing Research Program of Basic Research and Frontier Technology (cstc2019jcyj-msxmX0388), and Scientific and Technological Research Program of Chongqing Municipal Education Commission (KJQN201901614).

We have no competing interests to declare.

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
