## [Reviewer comments · Microbiology Spectrum]

Microbiology Spectrum

The zinc finger transcription factor BbCmr1 regulates conidium maturation in *Beauveria bassiana*

Jinfeng chen, Junjie Tan, Junyao Wang, Ajing Mao, Xueping Xu, Yan Zhang, XueLi Zheng, Yu Liu, Dan Jin, Xianbi Li, and Yanhua Fan

Corresponding Author(s): Yanhua Fan, Biotechnology Research Center, Southwest University, Beibei, Chongqing P.R. China

Review Timeline:

Submission Date:	November 2, 2021
Editorial Decision:	December 16, 2021
Revision Received:	December 23, 2021
Accepted:	January 11, 2022

Editor: Christina Cuomo

Reviewer(s): The reviewers have opted to remain anonymous.

Transaction Report:

DOI: <https://doi.org/10.1128/spectrum.02066-21>

December 16, 2021

Dr. Jinfeng chen
Biotechnology Research Center, Southwest University, Beibei, Chongqing P.R. China
Chongqing
China

Re: Spectrum02066-21 (The zinc finger transcription factor BbCmr1 regulates conidium maturation in *Beauveria bassiana*)

Dear Dr. Jinfeng chen:

Thank you for submitting your manuscript to Microbiology Spectrum. Two reviewers have provided feedback on your work that I would like you to address in a revision. When submitting the revised version of your paper, please provide (1) point-by-point responses to the issues raised by the reviewers as file type "Response to Reviewers," not in your cover letter, and (2) a PDF file that indicates the changes from the original submission (by highlighting or underlining the changes) as file type "Marked Up Manuscript - For Review Only". Please use this link to submit your revised manuscript - we strongly recommend that you submit your paper within the next 60 days or reach out to me. Detailed instructions on submitting your revised paper are below.

Link Not Available

Sincerely,

Christina Cuomo

Journals Department
Reviewer comments:

Reviewer #1 (Comments for the Author):

The authors report on a Zn²⁺Cys₆ transcription factor of *B. bassiana*, BbCmr1, that is part of the transcription factors in the central development pathway BrIA-AbaA-WetA in conidium maturation. Targeted gene knockout of Bbcmr1 resulted in conidial cell wall impaired, lower conidial germination rate under abiotic stresses. Further analysis found that Bbcmr1 positively regulated BbwetA expression, and BbBrIA positively regulated the expression of Bbcmr1, which forms a regulatory network mediating conidial development. This finding was critical to understand the molecular regulatory networks of conidial development in *B. bassiana* and provided avenues to engineer insect fungal pathogens with high-quality conidia. Overall, the data are presented in a logical manner, and the manuscript is well-written.

A few minor points are listed below.

- 1) Fig. 2, panel D, Fig. 3 panel C-E, Fig. 8, there are markings on the images to show significant difference between or among the test samples, either better images could be provided or the markings can be replaced.
- 2) Lin 175-198, author described that Bbcmr1 regulates the expression of genes involved in conidium maturation. Would recommend adding some comments on that Smr1, BrIA and WetA genes were also differentially expressed in Δ Bbcmr1 strain.

Reviewer #2 (Comments for the Author):

The study by Chen et al. focuses on the role of the zinc finger transcription factor BbCmr1 in conidium maturation of the entomopathogenic fungus *Beauveria bassiana*. The entire work is scientifically sound. The authors have done a nice job combining diverse cellular and molecular biology techniques to satisfactorily answer the research questions. As a general recommendation, try to avoid using excessive jargon. See below some minor comments.

line 52: "a fungal invertebrate pathogen", "an invertebrate pathogenic fungi" or "a fungal pathogen of invertebrates" reads better than "an invertebrate fungal pathogen"

line 58: cuticle-degrading hydrolases

Line 139 and Fig. 2 caption: please define CZA for the first time.

Line 193: to help a more fluent reading, please describe what is mCherry-tagged in line 193 and define EMSA in line 241.

Line 199: expression

Staff Comments:

Preparing Revision Guidelines

Please return the manuscript within 60 days; if you cannot complete the modification within this time period, please contact me. If you do not wish to modify the manuscript and prefer to submit it to another journal, please notify me of your decision immediately so that the manuscript may be formally withdrawn from consideration by Microbiology Spectrum.

Comments to the Author

The authors report on a Zn₂Cys₆ transcription factor of *B. bassiana*, *BbCmr1*, that is part of the transcription factors in the central development pathway BrlA-AbaA-WetA in conidium maturation. Targeted gene knockout of *Bbcmr1* resulted in conidial cell wall impaired, lower conidial germination rate under abiotic stresses. Further analysis found that *Bbcmr1* positively regulated *BbwetA* expression, and *BbBrlA* positively regulated the expression of *Bbcmr1*, which forms a regulatory network mediating conidial development. This finding was critical to understand the molecular regulatory networks of conidial development in *B. bassiana* and provided avenues to engineer insect fungal pathogens with high-quality conidia.

Overall, the data are presented in a logical manner, and the manuscript is well-written.

A few minor points are listed below.

- 1) Fig. 2, panel D, Fig. 3 panel C-E, Fig. 8, there are markings on the images to show significant difference between or among the test samples, either better images could be provided or the markings can be replaced.
- 2) Lin 175-198, author described that *Bbcmr1* regulates the expression of genes involved in conidium maturation. Would recommend adding some comments on that *Smr1*, *BrlA* and *WetA* genes were also differentially expressed in $\Delta Bbcmr1$ strain.

Christina Cuomo

Dear Dr. Christina Cuomo

We are grateful to you and the two reviewers for the constructive comments on our manuscript entitled "The zinc finger transcription factor BbCmr1 regulates conidium maturation in *Beauveria bassiana*". The manuscript has been revised based on the Reviewers' suggestions. We reorganized multipanel figures to be assembled into one file. Introduction, Results, Discussion, and Materials and Methods have been modified in response to specific reviewers' comments. We feel that the quality of our paper has been substantially improved after revision and appreciate your consideration of this work.

The complete reviewers' comments and a point-by-point responses are included in the revised submission.

Sincerely yours,

Yanhua Fan

Reviewer: 1

Comments to the Author

The authors report on a Zn²⁺Cys₆ transcription factor of *B. bassiana*, BbCmr1, that is part of the transcription factors in the central development pathway BrlA-AbaA-WetA in conidium maturation. Targeted gene knockout of Bbcmr1 resulted in conidial cell wall impaired, lower conidial germination rate under abiotic stresses. Further analysis found that Bbcmr1 positively regulated *BbwetA* expression, and BbBrlA positively regulated the expression of *Bbcmr1*, which forms a regulatory network mediating conidial development. This finding was critical to understand the molecular regulatory networks of conidial development in *B. bassiana* and provided avenues to engineer insect fungal pathogens with high-quality conidia.

Overall, the data are presented in a logical manner, and the manuscript is well-written.

Minor concerns:

1. Fig. 2, panel D, Fig. 3 panel C-E, Fig. 8, there are markings on the images to show significant difference between or among the test samples, either better images could be provided or the markings can be replaced.

Response: We thank the reviewer for the careful comment. We have provided better images and replaced the markings according to this suggestion, and the multipanel figures were assembled into one file.

2. Lin 175-198, author described that Bbcmr1 regulates the expression of genes involved in conidium maturation. Would recommend adding some comments on that Smr1, BrlA and WetA genes were also differentially expressed in Δ Bbcmr1 strain.

Response: In the revised version, we added some comments on that Smr1, BrlA and WetA genes were also differentially expressed in Δ Bbcmr1 strain (Line 199-201).

Reviewer: 2

Comments to the Author

The study by Chen et al. focuses on the role of the zinc finger transcription factor BbCmr1 in conidium maturation of the entomopathogenic fungus *Beauveria bassiana*. The entire work is scientifically sound. The authors have done a nice job combining diverse cellular and molecular biology techniques to satisfactorily answer the research questions. As a general recommendation, try to avoid using excessive jargon. See below some minor comments.

Minor concerns:

1. line 52: "a fungal invertebrate pathogen", "an invertebrate pathogenic fungi" or "a fungal pathogen of invertebrates" reads better than "an invertebrate fungal pathogen"

Response: Thank the reviewer for this suggestion. We have replaced the corresponding term in the introduction part (Line 52).

2. line 58: cuticle-degrading hydrolases

Response: The reviewer is right. The term of "cuticle hydrolases" was revised by "cuticle-degrading hydrolases" (Line 58-59).

3. Line 139 and Fig. 2 caption: please define CZA for the first time.

Response: We apologized for the ignorance of definition CZA for the first time. In the revised version, we define "CZA" as "Czapek-Dox Agar" (Line 139, Fig 2 caption).

4. Line 193: to help a more fluent reading, please describe what is mCherry-tagged in line 193 and define EMSA in line 241.

Response: This is a good suggestion. As suggested by the reviewer, we described “mCherry-tagged” (193-194) and EMSA (Line 244).

Line 199: expression.

Response: We have recognized the word of “expressions” was misused. In the current version, the mentioned mistake has been corrected (Line 202).

January 11, 2022

Dr. Yanhua Fan
Biotechnology Research Center, Southwest University, Beibei, Chongqing P.R. China
No.1 Tiansheng Rd
chongqing
China

Re: Spectrum02066-21R1 (The zinc finger transcription factor BbCmr1 regulates conidium maturation in *Beauveria bassiana*)

Dear Dr. Yanhua Fan:

Your manuscript has been accepted, and I am forwarding it to the ASM Journals Department for publication. You will be notified when your proofs are ready to be viewed. To ensure there are no delays in publication of your article, please ensure that the sequence data (PRJNA777036) is promptly released.

Sincerely,

Christina Cuomo
Editor, Microbiology Spectrum
